# Signaling Pathways Involved in the Development of Bronchopulmonary Dysplasia and Pulmonary Hypertension

**DOI:** 10.3390/children7080100

**Published:** 2020-08-18

**Authors:** Rajamma Mathew

**Affiliations:** 1Departments of Pediatrics, New York Medical College, Valhalla, NY 10595, USA; rmathew@nymc.edu; 2Departments of Physiology, New York Medical College, Valhalla, NY 10595, USA

**Keywords:** BPD (bronchopulmonary dysplasia), pulmonary hypertension (PH), inflammation, dysregulated signaling pathways, mesenchymal stem cells (MSC)

## Abstract

The alveolar and vascular developmental arrest in the premature infants poses a major problem in the management of these infants. Although, with the current management, the survival rate has improved in these infants, but bronchopulmonary dysplasia (BPD) is a serious complication associated with a high mortality rate. During the neonatal developmental period, these infants are vulnerable to stress. Hypoxia, hyperoxia, and ventilation injury lead to oxidative and inflammatory stress, which induce further damage in the lung alveoli and vasculature. Development of pulmonary hypertension (PH) in infants with BPD worsens the prognosis. Despite considerable progress in the management of premature infants, therapy to prevent BPD is not yet available. Animal experiments have shown deregulation of multiple signaling factors such as transforming growth factorβ (TGFβ), connective tissue growth factor (CTGF), fibroblast growth factor 10 (FGF10), vascular endothelial growth factor (VEGF), caveolin-1, wingless & Int-1 (WNT)/β-catenin, and elastin in the pathogenesis of BPD. This article reviews the signaling pathways entailed in the pathogenesis of BPD associated with PH and the possible management.

## 1. Introduction

The supplemental O_2_ need in premature infants at 36 weeks post-menstrual stage is defined as bronchopulmonary dysplasia (BPD), a chronic lung disease [1]. BPD occurs in infants with respiratory distress syndrome receiving positive pressure ventilation and O_2_ therapy. Antenatal steroid therapy, surfactant administration, and non-invasive ventilation have improved the survival rate in these infants. However, lung alveolar and vascular growth arrest place them at a risk of developing BPD. Infants born between 23–28 weeks of gestation are at a high risk of developing BPD, as the lungs at this stage evolve from canalicular to saccular stage. Under normal conditions, terminal saccules develop into alveolar ducts [2]. BPD begins as altered lung development even before delivery because of chorioamnionitis and tobacco exposure and is further complicated by postnatal exposure to O_2_, mechanical ventilation, and infection. The injury leads to the activation of inflammatory pathways [3]. Prenatal factors such as low gestational age, low birth weight for gestational age, and male gender are known predictors of the progression of chronic respiratory insufficiency in these infants associated with a higher mortality rate [4].

Gender appears to play a significant role both in the healthy and diseased lungs. Sex hormones exert regulatory effects on pulmonary pathophysiology. Importantly, surfactant appears in female neonatal lungs earlier than males, which would favor patency of small airways, thus contributing to higher airflow rate and lower airway resistance [5]. In addition, male mice exposed to hyperoxia exhibit decreased expression of angiogenesis markers, such as platelet endothelial cellular adhesion molecule (PECAM)1 and vascular endothelial growth factor receptor (VEGFR)2, and reduced nuclear factor κB (NF-κB) pathway activation in the lungs compared with female mice [6]. In normoxia, significantly higher cell migration and greater sprouting capability are observed in pulmonary microvascular endothelial cells from human female compared with male human endothelial cells. Furthermore, exposure to hyperoxia significantly reduces cell viability and proliferation in male pulmonary microvascular endothelial cells, but in female endothelial cells, the viability is maintained [7]. Moderate and severe BPD are significantly more common in male infants (63.3%) compared with female infants (36.6%); however, in infants with gestational age of 22–25 weeks, female gender is not a protective factor [8]. Fulton et al. [9] reported that the tracheal aspirate mesenchymal stromal cells (MSCs) from male infants developing BPD exhibited significantly lower messenger RNA (mRNA) expression of proliferative and anti-apoptotic factors, such as platelet derived growth factor receptor A (PDGFRA), fibroblast growth factor 7 (FGF7), wingless-type family member 2 (WNT2), sprouty 1 (SPRY1), matrix metalloproteinase 3 (MMP3), and forkhead box F2 (FOXF2). Furthermore, infants with associated pulmonary hypertension (PH) revealed severe BPD. In addition, they had undergone longer durations of O_2_ therapy, conventional or high frequency ventilation, and hospitalization. Oligohydramnios is reported to be a specific risk factor for PH in preterm infants with moderate or severe BPD [10]. Furthermore, poor in utero growth and postnatal growth restriction during the first weeks of life are associated with increased risk for BPD and PH [11]. Bhat et al. [12] reported the incidence of PH to be 17.9% in a series of 145 extremely low-birth-weight-infants. In addition, infants with PH were more likely to have had received O_2_ on day 28. Importantly, early detection of PH (within 14 days) in these low-birth-weight infants is associated with moderate to severe BPD and increased mortality rate [13]. The survival rate in infants with PH complicating BPD is around 53% at 2 years [14].

Lung morphogenesis is a highly orchestrated process involving a number of signaling pathways. A number of growth factors, microRNAs, transcription factors, and their associated signaling cascades regulate cellular proliferation, migration, survival, and differentiation during the formation of the peripheral lung in a well-orchestrated manner. The timing and the amount of expression of these signaling pathways are of paramount importance for the normal lung development. The pulmonary vasculature develops in close proximity to epithelial progenitor cells, regulated by interactions between the developing epithelium and underlying mesenchyme. Vessel formation in the lung takes place via two mechanisms: Proximal pulmonary arteries develop via angiogenesis and distal, smaller vessels form by vasculogenesis. The proximal and distal vessels connect to establish the final vascular network. Furthermore, lung vascular development directly corresponds to overall lung growth [15]. Throughout the lung developmental phase, a close relationship between blood vessels and airways exists. Experimental studies in mice have shown that during the development of alveoli, type II pneumocytes produce vascular endothelial growth factor (VEGF) mRNA that promotes angiogenesis, and the absence of VEGF results in poor alveolarization and reduced capillary number [16]. Importantly, in infants with BPD, reduced VEGF mRNA and protein expression are accompanied by a decreased Flt-1 and Tie-2 expression, indicating a significant abnormality in the pulmonary vasculature [17]. Furthermore, in the absence of lung epithelial cells, pulmonary vascular cells fail to proliferate, indicating that for the normal development of the pulmonary circulation, the tissue interactions are critical [18]. Thus, a close coordination between airways and vessels is crucial for normal lung development. Postnatal treatment with the inhibitor of angiogenesis results in reduced lung vascular density, alveolarization, and lung weight, further underscoring the important relationship between vascular growth and lung alveolar development [19].

## 2. Inflammation, Oxidant Injury

Chorioamnionitis and in utero exposure to cytokines lead to an inflammatory response that is likely to be associated with aberrant wound healing in the lungs. A rapid buildup of inflammatory cells and mediators resulting from an inflammatory response has a direct effect on the integrity of the alveolar capillary unit. Thus, an essential component of BPD pathogenesis is an imbalance between pro- and anti-inflammatory mediators that favors pro-inflammatory mechanisms [20]. It has further been shown that within 2 days of intra-amniotic endotoxin injection, the expression of cytokines, such as interferon-γ-inducible protein (IP)-10 and transforming growth factor (TGF)-β, known to inhibit vascular development increases, accompanied by a decrease in endothelial nitric oxide synthase (eNOS) in the small, fetal pulmonary vessels. In addition, the expression of other vascular markers such as VEGF mRNA and protein, VEGF receptor-2, and PECAM-1 decreases, accompanied by hypertrophy of smooth muscle cells (SMCs) of the distal pulmonary arterioles by day 7. Thus, the antenatal inflammatory exposures result in the postnatal inflammation and dysregulation of the lung development [21]. Importantly, amniotic fluid concentrations of interleukin (IL)-6, IL-1β, IL-8 and tumor necrosis factor (TNF)-α were reported to be significantly higher in mothers whose infants had BPD [22].

During polymorphonuclear leukocyte (PMN) degranulation, PMN-derived exosomes (CD63^+^/CD66b^+^) acquire surface-bound neutrophils elastase (NE) oriented in a configuration that is resistant to α1-antitrypsin. These exosomes bind and degrade extracellular matrix (ECM) through the integrin Mac-1 and NE, causing characteristic chronic obstructive pulmonary disease. Similar findings of the ECM remodeling were observed in BPD [23]. In these infants, infection, inflammation, oxygen toxicity, and ventilation-induced volume and baro-trauma, together with other factors, affect the postnatal maturation of the lung, leading to blunted alveolarization, dysmorphic pulmonary vasculature and PH [24]. A subtype of endothelial progenitor cells (EPCs), known as endothelial colony-forming cells (ECFCs), displays strong clonal proliferative potential capable of forming durable and functional blood vessels in animal models. Preterm ECFCs emerge in increased numbers as well as proliferate more rapidly. In addition, they differentiate into terminally differentiated endothelial cells (EC), but they are more susceptible to hyperoxia compared with term ECFCs. Antioxidants protect preterm ECFCs from hyperoxia, and highly proliferative ECFCs may participate in vascular repair [25].

## 3. Deregulated Signaling Pathways

### 3.1. Angiopoitins, Endostatin

An imbalance between pro- and anti-angiogenic factors triggered by inflammation resulting in disrupted angiogenesis leads to the development of PH in BPD. Angiopoietin-1 (Ang-1), an angiogenic mediator, is the primary agonist of the tyrosine kinase receptor (Tie) 2, and the effect of Ang-1/Tie2 signaling is context-dependent. Ang-1 has chemotactic and mitogenic effects on endothelial cells (ECs), and it inhibits apoptosis. In addition, it supports the localization of adhesion molecules in endothelial intercellular junctions, thus stabilizing blood vessels. Several cell types, such as ECs, SMCs, fibroblasts, epithelial cells, monocytes, neutrophils, and eosinophils, express Tie2 receptor. Chemotactic effects induced by Ang-1/Tie2 signaling lead to differentiation of mesenchymal cells to SMCs, and play a key role in maintaining the integrity of mature quiescent vasculature. Furthermore, in a murine model, loss of either Ang-1 or Tie2 is reported to be associated with severe microvascular defects and embryonic mortality [26]. Tie 2 activation leads to the suppression of TNF-α-stimulated leukocyte transmigration across endothelial monolayer, providing anti-inflammatory effects on ECs. In addition, Tie2 stimulation inhibits the expression of the NF-κB-responsive genes such as intercellular adhesion molecule-1 (ICAM-1), vascular cell adhesion molecule-1 (VCAM-1), and VEGF-induced E-selectin and tissue factor induced by TNF-α and VEGF [27]. Ang-1/Tie2 interaction inhibits NF-κB, resulting in a reduced transcription of pro-inflammatory mediators. Endostatin is activated by proteolytic cleavage from its precursor collagen XVIII. It has inhibitory effects on EC proliferation, migration, and tube formation. In addition, endostatin downregulates endothelial signaling cascades associated with pro-angiogenic activity [28].

During the development of lungs, endostatin plays an important role in angiogenesis. Together with pro-angiogenic growth factors, such VEGF-A, it guides the developing vasculature. In term infants, the circulating endostatin levels are higher compared with very-low-birth-weight (VLBW) infants, which indicates a temporal pattern of endostatin expression in fetuses. Furthermore, a high endostatin level in cord plasma is a predictor of the development of BPD in these infants [29]. Ang-1 stabilizes new blood vessels, whereas Ang-2 destabilizes ECs via Tie-2 receptor, enabling vascular sprouting. The increased levels of Ang-2 in airway fluid from infants with BPD and small-for-gestational-age infants indicate a link between fetal pulmonary and disrupted placental angiogenesis. The tracheal aspiration fluid from ventilated VLBW infants on day 10 displaying increased Ang-2 levels is associated with moderate BPD or death. In addition, during early postnatal days in the infants who developed mild to moderate BPD or died revealed a reduced ratio of Ang-1 to Ang-2 in tracheal aspirate fluid. Thus, the imbalance between Ang-1 and Ang-2 in airway fluid is indicative of a continued disturbance of alveolar and pulmonary vascular development in ventilated very preterm infants who develop BPD or die [30]. Ang-1 and Ang-2 both have binding sites on Tie2 and bind with similar affinity; and transgenic overexpression of Ang-2 displays vascular defects similar to what have been observed in Ang-1 or Tie2 deficiency [26]. These results indicate that an imbalance between pro-angiogenic and anti-angiogenic factors contribute to the impaired angiogenesis observed in BPD.

### 3.2. Transforming Growth Factor (TGF)-β

Multiple pathways, including TGF-β pathway, orchestrate lung development. A balanced and timed expression of TGF-β is essential for embryonic and fetal lung development. At the beginning of lung development, endogenous retinoic acid controls TGF signaling in the prospective lung field of the foregut that allows fibroblast growth factor (FGF) 10 expression and the induction of primary lung buds [31]. TGF-β1 overexpression during the critical period of postnatal rat lung alveolarization gives rise to morphological, pathological, and biochemical changes consistent with those seen in human BPD [32]. TGF-β overexpression during later period of lung development inhibits branching morphogenesis and alveolarization. It functions through downstream mediators, such as connective tissue growth factor (CTGF) and caveolin-1. An increase in TGF-β signaling is accompanied by a decrease in the expression of caveolin-1, a structural component of caveolae known to promote the degradation of TGF-β receptors [33]. In a mouse BPD model, hyperoxia is reported to significantly affect the TGF-β/bone morphogenetic protein (BMP) signaling in the lung and processes necessary for septation and alveolarization. Interestingly, Smad3 knockout mice between 7 and 28 days exhibit retarded alveolarization indicating that TGF-β also functions as a positive regulator of septation. Furthermore, in adult mice, Smad3 deficiency leads to enlarged airspaces and centrilobar emphysema in late life, suggesting an important role for TGF-β signaling in both the formation of alveoli and the maintenance of alveolar structure. Signaling by the TGF-β/BMP superfamily plays a pivotal role in lung development [34].

### 3.3. Caveolin-1

Caveolae (size 50–100 nm), nonclathrine-coated plasma membrane vesicles, are enriched in sphingomyelin, glycoshingolipids, cholesterol, and lipid-anchored membrane proteins. They form a salient signaling platform that compartmentalizes and integrates a number of signaling molecules and allow cross talk between different signaling pathways and mediate and integrate signaling events at the cell surface [35]. Caveolin-1, a major protein (mol wt. 22 kDa) constituent of caveolae, not only maintains the shape of caveolae, but also, through the caveolin-1 scaffolding domain (CSD, residue 82–101), interacts with proteins within caveolae. It regulates and stabilizes a number of proteins including Src family of kinases, endothelial NO synthase (eNOS), guanine nucleotide-binding (G) proteins (α-subunits), G protein-coupled receptors, H-Ras, protein kinase C (PKC), integrins, epidermal growth factor (EGF) receptor in an inhibitory conformation [36]. Importantly, after antenatal inflammation, caveolin-1 mRNA and protein expression was found to be low in lung tissues. However, TGF-β1 levels increased markedly with antenatal inflammation-induced lung remodeling. In addition, low levels of caveolin-1 were associated with the increased phosphorylation of Smad2/3, Stat3, and Stat1. Thus, it is likely that low levels of caveolin-1 and associated alterations in other signaling pathways contribute to BPD [37]. Caveolin-1 plays an important role in the function and homeostasis of the lungs after birth. Caveolin-1α, an early marker for lung vasculogenesis, is largely expressed in developing blood vessels. During postnatal period, caveolin-1 α is also expressed in alveolar Type 1 cells, in fully differentiated lungs [38]. Furthermore, increased caveolin-1 expression is a marker of the differentiation of lung alveolar epithelial type II cells into a type I phenotype, and the effects of dexamethasone, in part, are mediated by stabilization of caveolin-1 mRNA [39]. Caveolin-1, a marker of the mature, contractile SMC phenotype is essential for contractile protein expression induced by the growth factor TGF-β1.

In addition, caveolin-1 expression and caveolae number are highest in airway and vascular myocytes with a contractile phenotype. Thus, caveolin-1 plays key roles (both facilitative and repressive) in directing TGF-β1 signaling to specific intracellular pathways [40]. Caveolin-1 knockout mice that lack caveolae exhibit significantly reduced lung compliance, increased elastance, and airway resistance by three months of age. The decreased caveolin-1 levels accompanied by changes in other signaling pathways may have an important role in the pathogenesis of BPD [41]. In addition, antenatal exposure to lipopolysaccharide (LPS) results in decreased caveolin-1 mRNA and protein expression. Antenatal glucocorticoid prevents CTGF induction, caveolin-1 downregulation, and TGF-β signaling in fetal lungs [42]. The role of caveolin-1 in TGF-β signaling and TGF-β receptor internalization is quite important. The restoration of caveolin-1 function via cell permeable caveolin-1 scaffolding domain (CSD) has been shown to abolish spontaneous and TGF-β1-stimulated endothelium to mesenchymal transition (EndoMT) [43]. Caveolin-1, a known marker of the type I epithelial cell phenotype, plays a role in mechano-transduction of fetal type II epithelial cells. It functions as an inhibitory protein in stretch-induced type II cell differentiation via the extracellular signal-regulated kinase (ERK) pathway. However, in adult type II cells, caveolin-1 expression is relatively low. In contrast, in mice by embryonic day 16, both caveolin-1 and caveolin-2 are richly expressed in the developing lung parenchyma and within the epithelial cells that line the developing bronchioles [44]. In one study, infants with respiratory distress syndrome and PH revealed well-preserved expression of caveolin-1, PECAM-1, and von Willebrand factor (vWF), indicating that there was no disruption of the endothelial layer [45]. However, exposure to hypoxia leads to a tight complex formation between caveolin-1 and eNOS, rendering both molecules ineffective [46,47]. In two infants with BPD and associated inflammatory disease, the pulmonary arteries exhibited loss of endothelial caveolin-1 and PECAM-1, suggestive of endothelial membrane damage. An additional loss of vWF, indicative of extensive endothelial damage, was associated with enhanced expression of caveolin-1 in SMC, as has been reported in pulmonary arterial hypertension (PAH) [45]. Enhanced expression of caveolin-1 in SMCs, accompanied by a loss of caveolae, indicates that caveolin-1 is not in caveolae, but translocated to the cell membrane. Furthermore, this caveolin-1 in SMC is tyrosine phosphorylated, which is known to facilitate pathological conditions [48]. It is important to recognize that caveolin-1 function in noncaveolar site is distinctly different from caveolin-1 in caveolae [49]. Thus, not only the presence or the absence of caveolin-1 but also its location and its state are important factors in pathophysiology.

### 3.4. Connective Tissue Growth Factor (CTGF)

TGF-β1 contributes to normal lung development. However, TGF-β1 overexpression during critical period of lung alveoralization causes morphological changes observed in BPD. Downstream effecter of TGF-β1, CTGF can prolong wound healing and lead to fibrotic changes. TGF-β1 induces CTGF in fibroblasts and ECs. In sheep, endotoxin-induced chorioamniotic inflammation leads to increased TGF-β1 expression and reduction in CTGF. The decreased CTGF in EC may affect vascular development [50]. CTGF expression in EC is suggestive of its role in endothelial homeostasis and angiogenesis during embryonic development. Importantly, CTGF knockout mice exhibit vascular defects during embryogenesis [51]. CTGF, also known as CCN2, is necessary for normal lung development. Recent studies in experimental models have demonstrated the involvement of CTGF in the development of BPD, and the lung tissues from infants with BPD exhibit increased expression of CTGF. Increased CTGF expression induced by hyperoxia, inflammation, and mechanic ventilation may promote fibroblast proliferation, matrix production, and vascular remodeling. Overexpression of CTGF in alveolar epithelial type II cells disrupts alveolarization and vascular development, resulting in vascular remodeling and PH. Studies in a rodent model of hyperoxia-induced BPD have shown that inhibition of CTGF by a CTGF monoclonal antibody improved alveolarization and vascular development and decreased pulmonary vascular remodeling and PH [52]. Overexpression of CTGF induces β-catenin nuclear translocation that may play a role in the pathogenesis of BPD [53]. Furthermore, newborn rats exposed to hyperoxia for 14 days displayed the activation of β-catenin signaling, decreased alveolarization, and deregulated vascular development and PH. Treatment with CTGF antibody during hyperoxia prevented the activation of β-catenin signaling, improved alveolarization and vascular development, and reduced PH [54]. In addition, the CTGF overexpression promotes vascular SMC to express more extracellular matrix protein collagen I, fibronectin, increases proliferation, and migration, which could be reversed by an anti-CTGF antibody [55]. Thus, dysregulated CTGF in BPD appears to play an important role in vascular remodeling and PH.

### 3.5. Fibroblast Growth Factor 10 (FGF10)

A number of growth factors such as FGFs, bone morphogenetic protein (BMP)s, WNT, Sonic Hedgehog (SHH) is implicated in the developing lung morphogenesis. Several FGF ligands are expressed in the developing lungs, but for initial lung formation, only FGF10 is required [56]. FGF10 expressed by mesenchymal cells regulates branching morphogenesis during the early stages of lung development and continues to be expressed in the saccular stage. FGF10 promotes proliferation of the underlying epithelium and the induction and the extension of the epithelial bud. As the extension proceeds, the distal endoderm cells express high levels of FGF10, which induces BMP4 expression. The BMP4 acts a lateral inhibitor of budding, controlling FGF10 function, thus maintaining proper lung growth [57]. In vitro studies have further shown that FGF10 expressed by distal mesenchyme contributes to parabronchial SMC via BMP4 synthesis by epithelium. Thus, the regulation of BMP signaling appears to participate in fine-tuning SMC differentiation [58]. In addition, during embryonic development, mesenchymal cells expressing FGF10 are progenitors for airway and vascular SMC [59]. VEGFa is a target of FGF10 in developing lung epithelium and the reduction in FGF10 levels leads to decrease in VEGFa and vascular defect [60]. FGF10 is not only essential for epithelial progenitor cell proliferation but also for coordinated alveolar SMC formation and vascular development [61]. High levels of Bmp4 and SHH are expressed in the distal epithelium. FGF10 transcription is reduced in transgenic lungs over-expressing SHH in the endoderm, indicating that high levels of SHH downregulate FGF 10 [62]. Importantly, a reduction in FGF10 expression has been observed in the lungs of infants dying of BPD [63]. Furthermore, exogenous FGF-10 has been shown to reduce the inflammatory cytokines’ levels and reduced NF-κB p65 expression in mice lungs, indicating that FGF-10 attenuates hyperoxia-induced inflammation [64].

### 3.6. WNT/β-Catenin

For the early lung morphogenesis, the WNT/β-catenin signaling cascade is critical and it is an important pathway for self-renewal and differentiation of stem/progenitor cells. Zhang et al. [65] have examined canonical WNT/β-catenin signaling components in the human embryonic lungs at 7, 12, 17, and 21 weeks. Most of the canonical WNT signaling components were detected at 7 weeks that increased to high levels at 17 weeks followed by a decrease at 21 weeks. Furthermore, the expression of β-catenin in the respiratory epithelium of the embryonic lung is necessary for the growth and differentiation of peripheral epithelial cell progenitors. β-catenin deletion in the embryonic lung epithelial cells disrupts lung morphogenesis, restricting formation and differentiation of the peripheral lung and enhancing formation of the conducting airways [66]. Aberrant β-catenin signaling in response to acute and chronic lung injuries is associated with abnormal epithelial proliferation, fibroproliferative repair, and lung matrix remodeling. Both CTGF and fibronectin, the target genes of β-catenin, play an important role in extracellular matrix (ECM) deposition and in vascular remodeling. In addition, the inhibition of β-catenin signaling decreases hyperoxia-induced PH, right ventricular hypertrophy, pulmonary vascular remodeling, and the expression of CTGF and fibronectin [67]. Interestingly, unstimulated MSCs from infants developing BPD show higher phospho-glycogen synthase kinase (GSK)-3β, β-catenin, and α-actin contents, and phospho-GSK-3β and β-catenin both correlated with α-actin content [68]. TGF-β upregulates canonical WNT signaling and inhibits the peroxysome proliferator activated receptor gamma (PPARγ). The absence or a decrease in the WNT/β-catenin signaling during the canalicular stage of pulmonary development, partly related to inflammatory processes, severely affects the developmental processes during the subsequent saccular and alveolar stages. PPARγ stimulates transdifferentiation of myofibroblasts into lipofibroblasts, which helps normal alveolarization. Importantly, hypoxia and hyperoxia promote upregulation of the canonical WNT/β-catenin system as well as TGF-β accompanied by downregulation of PPARγ [69]. Interestingly, the administration of PPARγ agonist, rosiglitazone, has been shown to prevent hyperoxia-induced molecular and morphological changes in a rat model [70]. In addition, increased mesenchymal Wnt5A during the saccular-stage hyperoxia injury contributes to the impaired alveolarization and septal thickening in BPD. Wnt5A inhibition abrogates the BPD transcriptomic phenotype induced by hyperoxia [71].

### 3.7. Vascular Endothelial Growth Factor (VEGF)

During the period of alveolarization, the lung undergoes vascular growth involving two basic processes: Vasculogenesis, the formation of new blood vessels from endothelial cells within the mesenchyme, and angiogenesis, the formation of new blood vessels from sprouts of preexisting vessels. For normal lung development, coordination of distal air space and vascular growth is crucial, and angiogenesis is required for alveolarization [72]. Furthermore, VEGF is pivotal for vascular and parenchymal maturation and surfactant production [73]. Neonatal exposure to hyperoxia in rats causes abnormalities in the pulmonary alveolar and capillary structure, similar to what is seen in BPD [74]. In addition, VEGFR inhibitor Sugen 5416 treatment in rats leads to impaired alveolarization and pulmonary vascular growth and PH [75]. In two different studies with a rat model of BPD, intratracheal adenovirus-mediated VEGF gene therapy or intramuscular VEGF gene therapy improved survival, promoted lung capillary formation, and conserved alveolar development. Furthermore, VEGF gene transfer increased alveolar eNOS expression, indicating that the beneficial effect of VEGF might be, at least in part, NO mediated. In a similar study, treatment of newborn rats with a VEGF receptor inhibitor resulted in abnormal lung structure and PH [76,77]. Lungs of infants with BPD who died displayed the evidence of defective alveolar septation and capillary formation associated with reduced expression of VEGF and VEGF receptor 1 (VEGF-R1). Defective VEGF signaling and activation of TGFβ reduce the expression of VEGF-R2 in endothelial cells, which may contribute to the defective lung septation and angiogenesis observed after prolonged mechanical ventilation. Mechanical stretch, even without hyperoxia, is a major stimulus for apoptosis, leading to impaired alveolar septation and increased deposition and dispersion of lung elastin [78]. VEGFa is expressed mainly by alveolar type 1 (AT1) cells. Carbonic anhydrase 4 (Car4) ECs are separated from AT1 cells by a limited basement membrane without intervening pericytes. Epithelial VEGFa deletion leads to the loss of Car4 ECs. In the absence of Car4 ECs, despite the normal appearance of myofibroblasts, alveolar space is aberrantly enlarged. These observations indicate a signaling role of AT1 cells [79]. Importantly, overexpression of VEGF in newborn mice induces inducible nitric oxide synthase (iNOS) and eNOS-dependent lung simplification, pulmonary edema, and oxidant stress. In VEGF transgenic mice, NOS inhibition has been shown to decrease oxidative stress, vascular permeability, and angiogenesis [80]. These results show that timing and the correct amount of expression of VEGF and other factors are essential for normal alveolarization and angiogenesis.

### 3.8. MicroRNAs

MicroRNAs (miRs) are small, conserved, regulatory RNAs in mammals that account for about 1% of the genome and they regulate gene expression. Differential expressions of certain miRs participate in the different stages of alveolar development during the progression of BPD [81]. Studies in mice with conditional knockout of Dicer in lung epithelial cells have shown that it leads to epithelial branching failure, thus highlighting the essential regulatory role of miRs in lung epithelial morphogenesis [82].

A number of miRs and their targets are involved in normal lung alveolar septation, and it is likely that their deregulation contributes to hyperoxia-induced abnormal lung development. Recent studies have further implicated the involvement of miRs in hyperoxia-induced lung injury, including BPD.

Hypoxia inducible factor-1α (HIF-1α) plays a crucial role in postnatal lung development, particularly in recovery from hyperoxic injury. The expression of miR-30a that has pro-angiogenic, anti-inflammatory, and anti-fibrotic effects is decreased in human BPD. Hif-1α is thought to affect differential sex-specific miR-30a expression that may contribute to protection from hyperoxic lung injury in female neonatal mice through decreased Snai1 expression [83]. In addition, Alam et al. [84] have shown enhanced expression of miR199a-5p in hyperoxia-exposed mice lungs, endothelial and epithelial cells, and also in tracheal aspirates of infants developing BPD, accompanied by a significant reduction in the expression of its target, caveolin-1. The miR199a-5p-mimic increases inflammatory cells, cytokines, and lung vascular markers, leading to the worsening of hyperoxic acute lung injury. Furthermore, miR199a-5p-inhibitor treatment attenuates hyperoxic acute lung injury.

In addition, the lungs of neonatal mice exposed to hyperoxia display significantly increased levels of miR-34a; and inhibition or deletion of miR-34a improves the pulmonary phenotype and BPD-associated PH. Administration of Ang-1, a downstream target of miR34a, has been shown to ameliorate BPD and PH [85]. The expression of miR-154 is reported to increase during lung development and decrease during postnatal period. The regulation of miR-154 in postnatal lung is an important physiological switch that permits the induction of the correct alveolar developmental program. The failure of miR-154 downregulation leads to suppression of alveolarization, resulting in alveolar simplification; and hyperoxia exposure maintains high levels of miR-154 in alveolar type 2 cells (AT2). Importantly, caveolin-1 is a key downstream target of miR-154. Overexpression of miR-154 results in the downregulation of caveolin-1 protein associated with increased phosphorylation of Smad3 and TGF-β signaling. In addition, AT2 cells overexpressing miR-154 display decreased expression of AT2 markers and increased expression of AT1 markers [86].

Interestingly, the hyperoxia-induced inhibition of miR-489 is thought to be a poor attempt at maintaining alveolar septation during hyperoxic exposure [87]. The miRs in cluster 4 including miR-127 exhibit the highest expression during the late stage of fetal lung development; and miR-127 expression gradually shifts from mesenchymal cells to epithelial cells during the developmental progression. In fetal lung organ culture studies, the overexpression of miR-127 resulted in decreased terminal bud count and increased terminal and internal bud sizes, causing unevenness in bud sizes, indicative of improper development. Thus, miR-127 appears to have a critical role in fetal lung development [88]. In another study, Lal et al. [89] have shown higher number of exosomes released in the tracheal aspirate from infants with severe BPD compared with gestational age–matched controls. However, the miR 876-3p expression was reduced in infants with severe BPD as well as in an animal model of hyperoxia-induced BPD. Exosomal miR 876-3p expression progressively decreased in bronchoalveolar lavage fluid of hyperoxia-exposed pups. Gain of function of miR 876-3p improved the alveolar architecture in the *in-vivo* BPD model, thus indicating a link between miR 876-3p and BPD. These studies highlight the role of a number of miRs in the pathophysiology of BPD.

## 4. Loss of Barrier Function

In premature infants, hyperoxia exposure not only leads to alveolar arrest in the lungs but also impairs alveolar epithelial junctional integrity. Tight junctions are located at alveolar type I–type II cell interfaces and regulate para-cellular fluid permeability through the expression of claudins, a transmembrane family of proteins. In in-vitro studies, neonatal alveolar epithelial cells on exposure to hyperoxia have shown to exhibit increased para-cellular leak and significant reduction in the mRNA and protein levels of claudin 3 and in the mRNA levels of claudin 18 and claudin 5 [90]. Mizobuchi M. et al. [91] have shown 44% (total 54) of premature infants (<28 wks gestational age) requiring ventilatory support beyond one week developed severe leaky lung syndrome. Hydrocortisone therapy seemed to have helped. Importantly, human fetal lungs (23–24 weeks of gestational age) exhibit significantly lower levels of claudin 18. Claudin 18 knockout mice have barrier dysfunction, lung injury, and impaired alveolarization [92]. In addition, the expression of occludin and zonal occludens-1 (ZO-1) is reduced during hyperoxia-induced acute lung injury in neonatal animals leading to the disruption of epithelial tight junction barrier [93]. Furthermore, in response to oxidant stress, alveolar epithelial cells increase the expression of TGF-β, which is known to exacerbate the acute phase of lung injury and deregulate alveolar epithelial barrier function by promoting epithelial-to-mesenchyme cells’ transformation (EMT), resulting in the downregulation of the expression of tight junction proteins [94]. Interestingly, caveolin-1 colocalizes with occludin at tight junctions, in raft-like compartments, which may have a role in regulation of para-cellular permeability [95]. Importantly, a decrease in cavolin-1 mRNA and protein levels during hyperoxia has been reported in in vitro as well as in in-vivo studies. Caveolin-1 colocalizes with tight junction proteins in pulmonary epithelial cell and it negatively regulates inter-endothelial junctional permeability [33]. Furthermore, exposure to hyperoxia results in the downregulation of caveolin-1 gene transcription and protein expression that precede the downregulation of ZO-1, occludin, and claudin-4 expression at both the mRNA and protein levels; and caveolin-1 upregulation prevents the hyperoxia-induced pulmonary epithelial barrier destruction and tight junction protein loss [96]. Gap junctions at the plasma membrane levels provide direct cell–cell contact, which enables diffusion of soluble signaling molecules between cells, and maintain intercellular communication. Gap junction channels are composed of proteins known as connexins. Connexin40 is generally expressed in pulmonary vascular ECs. The loss of vascular connexins has a deleterious effect on lung architecture and remodeling, indicating that coordinated regulation of pulmonary epithelial and vascular compartments is essential for proper development and maintenance of lung structure [97]. Caveolin-1 appears to have a crucial role in lung morphogenesis, which needs to be explored further.

## 5. Aberrant Remodeling of Extracellular Matrix (ECM)

The ECM is an intricately integrated system of interacting molecules needed for the normal functioning of the lung. Elastic fibers are a major component of ECM necessary for the lung development and for ensuring the transmission of mechanical forces equally to all parts of the lung. Elastin is widely distributed in the mammalian lung compartments such as pleura, septa, large vessels, and elastic cartilage. The respiratory parenchyma has the highest concentration of elastin. Interstitial and inflammatory cells produce elastases. Matrix metalloproteinases (MMPs) secreted by mammalian cells have elastolytic activity [98]. During fetal development, lung elastic tissue maturation is tightly controlled [99]. Aberrant remodeling of ECM plays an important role in the pathogenesis of BPD. During lung development, the ECM components (laminin and elastin) interact with a variety of lung cells in a coordinated and dynamic manner, thus maintaining proper morphogenesis and maturation. Laminin participates in alveolar growth and alveolar capillary network. Dysregulation of laminin results in decreased capillary density and impaired distal epithelial/mesenchymal cell differentiation. Elastin fibers provide elastic recoil of the lungs during breathing [100]. Coordinated action of gene expression, cell–cell communication, physical forces, and cell interactions with the ECM guide the normal lung development. Perturbations of ECM structures, such as dysregulated collagen deposition and disturbed elastin fiber organization, are distinctive features of clinical and experimental BPD [101]. Cross-linking of collagen and elastin, a key step in ECM, imparts stability and capability to the ECM. In patients as well as in animal models of BPD, the function of ECM cross-linking enzymes is deregulated and alveolarization is blocked, indicating that perturbed ECM cross-linking impacts alveolarization [102].

The lysyl oxidase, belonging to the family of amine oxidases, catalyzes the covalent cross-linking of lysine and hydroxylysine residues in collagen and elastin, and promotes the ECM stability. Recent reports indicate that the expression and activity of lysyl oxidase are increased in the lungs of patients affected with BPD as well as in the rodent models of BPD [103]. The elevated lysyl oxidase activity in the pulmonary vasculature contributes to the persistence and overabundance of ECM components (collagen and elastin fibers), resulting in improper remodeling and vascular disease. These fibers probably are resistant to proteolytic degradation, which could disturb the intricate balance between matrix synthesis and degradation that accompany vascular injury. In addition, modulation of lung matrix cross-linking can affect pulmonary vascular remodeling associated with PH. Furthermore, the expression of lysyl oxidases has been shown to be dysregulated in both clinical PH (idiopathic PAH, PAH associated with ventricular septal defect, and chronic thromboembolic PH) and hypoxia-induced PH in mice [104]. Thus, the increased lysyl oxidase activity-induced excessive stabilization of the ECM might impede the normal matrix remodeling necessary for pulmonary alveolarization and thereby contribute to the pathological features of BPD [105]. These results indicate potential role for lysyl oxidases in normal lung development, as well as in perturbed late lung development associated with BPD.

An important part of lung interstitium is elastin, which facilitates recoil of the pulmonary vasculature, conducting airways, and airspaces. During the lung development, elastin plays a crucial role in the formation of alveoli and blood vessels. Elastic fibers provide the structural integrity and distensibility of airways, alveoli, blood vessels, and ECM; and they are produced during late fetal and neonatal stages of development [106]. During arterial development, elastin controls proliferation of SMCs and stabilizes arterial structure. Disruption of elastin results in subendothelial proliferation of SMCs, thus contributing to vascular obstructive disease [107]. In another study, newborn mice treated with mechanical ventilation and O_2_ for up to 24 h exhibited increased lung elastase activity, accompanied by elastic fiber degradation and remodeling of the ECM. These changes were associated with activation of TGF-β and increased apoptosis, resulting in defective formation of alveoli and pulmonary microvessels coupled with increased elastin synthesis, increased elastase activity, and reduction in proteins that regulate elastic fiber assembly. Furthermore, increased elastolytic activity occurring during mechanical ventilation with O_2_ leads to increased production of poorly organized elastic fibers that contribute to the abnormal lung structure and function observed in BPD [108]. In infants with BPD, plasma elastase levels were higher, without an increase of plasma elafin, an elastase inhibitor, compared to infants without BPD. In a mouse model, during mechanical ventilation and O_2,_ treatment with elafin resulted in the inhibition of elastase activity and associated inflammation, suppression of TGF-β signaling, thereby limiting apoptosis and alveolar disruption. However, elafin did not improve the adverse effects on the pro-angiogenic regulatory pathway. This was thought to have been related to the inhibition of NF-κB, resulting in impaired VEGF signaling and loss of pulmonary microvessels [109,110].

Tenascin-C is another ECM glycoprotein synthesized during embryonic development. It is absent or reduced in most adult tissues, but re-expressed during wound healing, inflammation, and in tumors. Its expression is regulated in time and space, with a strong expression occurring during specific morphogenetic events throughout embryogenesis and organogenesis. During early postnatal period, tenascin-C plays a role in the regulation of alveolarization/septation and microvascular maturation [111]. In another study, tenascin-C was shown to be greatly expressed in the remodeled fibrotic alveolar walls underneath regenerative epithelium. The cells in these locations showed SMC actin immunoreactivity, indicative of myofibroblast phenotype [112,113]. These results further underscore the fact that for proper lung morphogenesis, the timing and the amount of expression of signaling factors are of paramount importance.

## 6. Therapeutic Implications

Prematurity, mechanical ventilation, and O_2_ requirement result in the deregulation of a number signaling pathways leading to impaired alveolarization and angiogenesis. Non-invasive ventilation has proven to be a better alternative. The treatment with inhaled NO in BPD has not proven to be beneficial in premature infants. The use of inhaled prostacyclin (iloprost) has been shown to shorten the duration of mechanical ventilation and promote clinical improvement. Furthermore, high tidal volume ventilation increases lung cyclic guanosine monophosphate (cGMP) levels that lead to endothelial barrier dysfunction; in contrast, iloprost attenuates stretch-induced endothelial disruption, and protects barrier function. Once the infants are off the ventilator, treatment with sildenafil would be beneficial in order to facilitate angiogenesis [114]. However, as noted earlier, prenatal inflammation, prematurity, mechanical ventilation, and high oxygen lead to further inflammation and oxidant injury, resulting in deregulation of a number of signaling pathways. For therapy to be successful, one needs to address multiple deregulated signaling pathways.

Mesenchymal stem cells (MSC) are considered the best cell types for tissue regenerative therapy. They modulate a number of biological functions such as tissue repair and immune system and downregulate the inflammatory responses. MSC-derived exosomes have protective effects on ischemia-reperfusion injury via inhibition of inflammatory responses and cell apoptosis. Recent studies have shown that MSC-derived extracellular vesicles (EVs) containing miRs promote cell and tissue repair and regeneration [115]. Paracrine factors released by MSCs include the secretion of growth factors and cytokines including VEGF, FGF, stromal cell-derived factor-1, TGF-β, and IL-1 receptor antagonist [116]. The most common human adult tissue sources for MSCs are bone marrow, adipose tissue, umbilical cord tissue, and placenta. MSCs interact with their neighboring cells and contribute cell-based responses, which could be therapeutic [117]. Interestingly, the treatment with human amniotic fluid stem cells in adult hyperoxia-exposed rats resulted in reduced expression of IL-6 and IL-1β as well as IFN-γ and TGF-1β in lung tissues accompanied by an improvement in the VEGF expression [118]. In another study, human amniotic fluid stem cells with upregulated VEGF expression had a superior effect on lung injury in preterm rabbits, whereas naïve human amniotic fluid stem cells showed a moderate therapeutic potential [119]. Importantly, female bone marrow-derived MSC exhibit greater therapeutic efficacy in reducing neonatal hyperoxia-induced lung inflammation and vascular remodeling in rats. Furthermore, the beneficial effects of female MSC were more pronounced in male animals. These results indicate that female MSC could be the most potent bone marrow-derived MSC population for lung repair in severe BPD associated with PH [120]. In addition, female MSC displayed significantly greater recovery of left ventricular developed pressure in rat hearts with ischemia-reperfusion injury compared with male MSC-treated hearts. Male MSCs produce significantly greater TNF-α and lesser VEGF than female MSCs [121]. Chang et al. [122] reported in 2014 treatment of nine preterm infants (gestational age 25.3 ± 0.9 weeks) treated with intra-tracheal transplantation of human umbilical cord blood-derived MSCs. No adverse effects were observed at 7 days post-treatment; the severity of BPD was noted to be low. Furthermore, the levels of IL-6, IL-8, MMP9, TNFα, and TGFβ were lower in tracheal aspirates of these infants. These studies showed beneficial effects of treatment with MSCs on lung development. However, a longer follow-up is necessary.

Interestingly, MSC-derived EVs, but not fibroblast-derived EVs, were equally effective as parental MSCs in attenuating H_2_O_2_-induced cell death and in abrogating impaired alveolarization, angiogenesis, and the anti-inflammatory and anti-apoptotic effects. These effects were eliminated by the VEGF-knockdown MSC-derived EV transplantation. This indicates that the VEGF present within the MSC-derived EVs is a critical paracrine factor that plays an important role in reducing hyperoxic lung injuries in newborn rats [123]. Recent studies have established MSC-derived EVs, especially exosomes, as one of the main therapeutic vectors of MSCs. MSC-derived EVs mimic the function of parental MSCs by transferring their components such as proteins/peptides, lipids, DNA, mRNA, miRNA, and organelles to recipient cells. Intra-tracheal-administered MSC-EVs appeared to be more effective than MSC in improving BPD-associated abnormal alveolarization and pulmonary vascular remodeling [124]. In another study, exosomes isolated from media conditioned by human MSC cultures were used to treat hyperoxia-exposed newborn mice. MSC-exosome treatment resulted in improved lung function, mitigation of BPD, decreased fibrosis, and amelioration of pulmonary vascular remodeling and PH. In addition, mechanism of action of MSC-exosome was considered to be associated with modulation of lung macrophage phenotype [125].

In summary, BPD is a major cause of neonatal morbidity and mortality. As shown in Figure 1, antenatal inflammation, prematurity, mechanical ventilation, and O2 requirement resulting in volume and baro-trauma lead to the disruption of highly orchestrated function of a number of signaling pathways required for normal morphogenesis. Deregulated repair mechanism results in adverse effects on vascular and alveolar development. Importantly, preterm birth itself has an increased risk of developing PH in children and adults even after adjusting for known risk factors such as chronic lung disease, congenital diaphragmatic hernia, chromosomal abnormalities, and congenital heart defects [126]. The experimental data on the use of MSC and MSC-derived EVs in BPD are quite encouraging. It is of interest that female MSCs produce less TNF-α and increased VEGF and have been proven to be of superior therapeutic value in cardiovascular and lung diseases. It appears that the treatment with MSC-EVs (especially genetically modified) may have an advantage over cell therapy. However, more studies are necessary to establish the benefits of MSC-EV therapy.

## Figures and Tables

**Figure 1 children-07-00100-f001:**
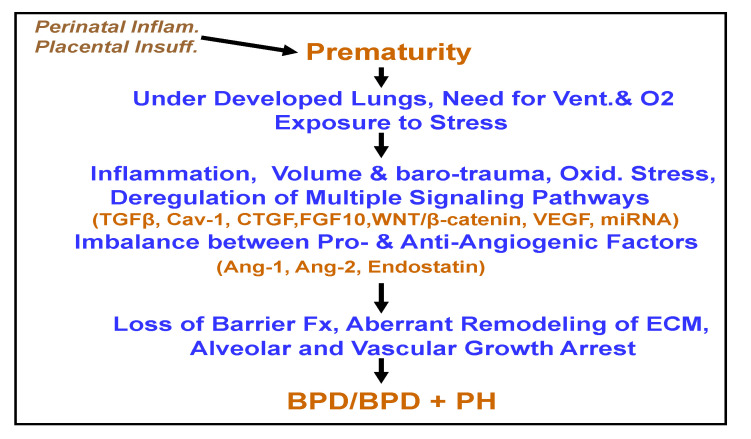
This figure recapitulates the development of bronchopulmonary dysplasia (BPD)/BPD+ pulmonary hypertension (PH) in premature infants. Ang-1 = angiopoietin-1, Ang-2 = angiopoietin-2, Barrier Fx = barrier function, ECM = extracellular matrix, Oxid= oxidative, Placental Insuff = placental insufficiency, Perinatal Inflam = perinatal inflammation, Vent = ventilation.

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
