# Peer review of "Signaling Pathways Involved in the Development of Bronchopulmonary Dysplasia and Pulmonary Hypertension"

_children, 2020, doi:10.3390/children7080100_

Round 1

Reviewer 1 Report

This is a well written review on the metabolic pathways involved in the development of Bronchopulmonary dysplasia.

As this review is almost focused on metabolic pathways, I would suggest to modify the title. I suggest a title that clarifies the aim of the review, maybe "Signaling pathways involved in the development of BPD and PH...."

Line 105-112: it is not clear if ECFC are protective or not... the author should better specify their role.

The paper should be checked for spelling errors (for example line 50 mesenchymal, line 124 murine model, line 139 endostatin level and so on)

Line 15: check the sentence, I think a word is missing

line 328: correct "micro RNAs"

References:

In ref 1 the title of the paper in missing

Some authors in the references are underlined. Please check this section and remove underlines

Author Response

Reviewer #1:

This is a well written review on the metabolic pathways involved in the development of BPD. Thank you for your kind remark.

As this review is almost focused on metabolic pathways, I would suggest to modify the title. I suggest a title that clarifies the aim of the review, maybe “Signaling pathways involved in the development of BPD and PH.

I agree with the suggestion. The title you have suggested appears appropriate for this review. I have changed the title.

Line 105-112: It is not clear if ECFC are protective or not… the author should better specify their role.

ECFC are thought to be involved in normal vascular development and in physiological responses to vascular injury. The statement has been modified: “In addition, they differentiate into terminally differentiated endothelial cells (EC), but they are more susceptible to hyperoxia compared with term ECFCs” (Lines 112-113)

The paper should be checked for spelling errors (for example line 50 mesenchymal, line 124 murine model, line 139 endostatin level and so on).

I apologize for the spelling errors. Typing is not my forte, and the computer itself introduces errors. I have carefully gone through the manuscript and found several typographical errors, which I have corrected. I hope I have corrected all the errors.

Line 15: Check the sentence; I think a word is missing. This statement has been corrected. The sentence appears as: “Despite considerable progress in the management of premature infants, therapy to prevent BPD is not yet available.

Line 328: Correct “microRNAS”. The typographical error has been corrected.

References: In reference 1, the title of the paper is missing. The title of the ref #1 has been inserted.

Some authors in the references are underlined. Please check this section and remove underlines. I hope I have succeeded in removing all the underlines, which had appeared at several places.

Reviewer 2 Report

Author have written a good and thorough review on BPD and pulmonary hypertension. Although, the title sounds more like a clinical review, this is more like a basic science review and may be better to make slight modification in title to reflect that.  It would also be helpful to add some tables or figures for easy reading of the content.  

Otherwise, this is a great detailed review of the topic and will be a good addition to the literature. 

Author Response

Author has written a good and thorough review on BPD and pulmonary hypertension. Although the title sounds more like a clinical review, this is more like a basic science review and may be better to make a slight modification in title to reflect that. It would also be helpful to add some tables or figures for easy reading of the content. Otherwise this is a great detailed review of the topic and will be good addition to the literature.

Thank you for your comments. I have changed the title to “Signaling pathways involved in the development of bronchopulmonary dysplasia and pulmonary hypertension”. I have added a Table describing major events during the progression of BPD/BPD+PH. I hope it is reasonable.